# Potential Impact of Climate Change on Schistosomiasis: A Global Assessment Attempt

**DOI:** 10.3390/tropicalmed3040117

**Published:** 2018-11-03

**Authors:** Guo-Jing Yang, Robert Bergquist

**Affiliations:** 1Department of Epidemiology and Public Health, Swiss Tropical and Public Health Institute, Socinstrasse 57, CH-4002 Basel, Switzerland; 2University of Basel, CH-4002 Basel, Switzerland; 3Ingerod, SE-454 94 Brastad, Sweden; robert.bergquist@outlook.com

**Keywords:** climate change, schistosomiasis, distribution, intermediate snail host, transmission, modelling

## Abstract

Based on an ensemble of global circulation models (GCMs), four representative concentration pathways (RCPs) and several ongoing and planned Coupled Model Intercomparison Projects (CMIPs), the Intergovernmental Panel on Climate Change (IPCC) predicts that global, average temperatures will increase by at least 1.5 °C in the near future and more by the end of the century if greenhouse gases (GHGs) emissions are not genuinely tempered. While the RCPs are indicative of various amounts of GHGs in the atmosphere the CMIPs are designed to improve the workings of the GCMs. We chose RCP4.5 which represented a medium GHG emission increase and CMIP5, the most recently completed CMIP phase. Combining this meteorological model with a biological counterpart model accounted for replication and survival of the snail intermediate host as well as maturation of the parasite stage inside the snail at different ambient temperatures. The potential geographical distribution of the three main schistosome species: *Schistosoma japonicum*, *S. mansoni* and *S. haematobium* was investigated with reference to their different transmission capabilities at the monthly mean temperature, the maximum temperature of the warmest month(s) and the minimum temperature of the coldest month(s). The set of six maps representing the predicted situations in 2021–2050 and 2071–2100 for each species mainly showed increased transmission areas for all three species but they also left room for potential shrinkages in certain areas.

## 1. Introduction

Schistosomiasis, caused by trematode parasites with a predilection for intestinal and urogenital venous circulation in the human definitive host, is one of the neglected tropical diseases (NTDs) selected for increased attention by the World Health Organization (WHO) [1]. Six different species of *Schistosoma* are capable of infecting humans, each depending on a certain snail species as an intermediate host. Humans infect snails by depositing parasite eggs (excreted in feces or urine) in waterlogged areas and humans are infected or reinfected when in contact with water containing schistosome cercariae released from infected snails. The disease is generally chronic, although schistosomiasis is often a contributing factor to premature death, direct mortality is comparatively low. Transmission of schistosomiasis has been reported in 78 countries and more than 800 million people in Africa, Latin America, the Middle East and Southeast Asia live in areas endemic to schistosomiasis [2], with up to 250 million actually infected [3]. Chemotherapy in amounts sufficient for more than 100 million school age children per year has been pledged by the private sector and development partners, but there is still a large discrepancy between the number of people requiring preventive treatment and those actually receiving it [4]. In addition, the number of people suffering from this infection continues to rise as a reflection of ongoing population growth, which is particularly high in endemic areas. 

Schistosomiasis is not as neglected as many other tropical diseases since it has a large research focus and still remains one of the most prevalent infections in the world, estimated to correspond to 4.5 million disability-adjusted life years (DALYs) by the WHO Expert Committee in 2002 [5]. However, this estimate was based on a wider range of pathologies than that used in the global burden of disease (GBD) study of 1990 [6]. It was also higher than the updated GBD in 2010 [7] that put the burden of schistosomiasis at 3.3 million DALYs. Importantly, however, the GBD figure of 2010 was high enough to put this disease as no. 3 after malaria and tuberculosis on the NTD list [3]. This increase was due to the inclusion of diarrhea, dysuria and anemia in the DALY score which was not counted before. While later updates [8] show a sharply lower DALY score for schistosomiasis, other authors [9,10,11] hold that the true impact of this disease is several-fold higher because of the low weight given by the GBD estimates to subtle symptoms and pathology in individuals with infections too light to be revealed by diagnostics based on parasite egg-detection. Indeed, high-definition circulating antigen tests, such as the point-of-care circulating cathodic antigen (POC-CCA) assay indicates that the current number of infections may be at least 10 times higher than that shown by egg-detection [12].

The epidemiology of schistosomiasis (and that of all other organisms) must be seen in the light of the perceived ongoing climate change. The latest half century has seen signs of global warming, mainly thought to be due to the burning of coal and oil at an increasingly large scale. From 1990, reports relevant for our understanding of climate change, including options for its mitigation, are regularly produced by the Intergovernmental Panel on Climate Change (IPCC) [13] for the United Nations Framework Convention on Climate Change (UNFCCC). IPCC bases its assessments on the published, scientific literature and opinions of invited independent researchers. An important part of its work is to do with global circulation models (GCMs), which are currently used to predict the climate for the next 80 years based on complex mathematical representations of the Earth’s energy balance between atmosphere, total land mass, sea and ice cover. These components interact as a coupled system, whose status emerges from equations based on the dynamic values of various climate variables, e.g., temperature, winds, etc., at each point on the globe. Climate modelling uses current and historic data to attempt the prediction of future climate scenarios from the present time to the end of the 21st century. 

The projections of the GCMs disagree due to various forms of natural variability included in the models. Fortuitously, this variability can be reduced by averaging an ensemble of simulations, resulting in universal agreement. Generally, such averaged model ensembles produce simulations of current and past large-scale climates that agree with observation. Further confidence comes from the fact that converging GCMs also produce an accurate ‘hindcast’ of previous climate change that took place in the 1900s. Evidence is already available in the form of a rising average global temperature and amplified warming of air and oceans, particularly in the Arctic, leading to rising sea levels, dry places becoming dryer and wet places wetter. 

The fact, temperature changes impact snail distributions in general as well as the maturation of the intermediate stages of the parasite inside this intermediate host, makes a discussion of future changes in the distribution of schistosomiasis complex. However, it is useful to know that water temperature below freezing puts an absolute limit to snail survival. Indeed, the climate-dependent, long-term (as opposed to seasonal) movements of the ‘frost line’ at northern latitudes indicates a diffuse zone north of which schistosomiasis transmission cannot occur. Immediately south of this zone, transmission is governed by the prevailing temperature and the time it stays above a certain limit, a fact rooted in the relationship between the development of plants and the ambient temperature first mentioned in the 1700s by Réaumur who coined the ‘degree-day unit’ and used it as a measure of crop maturation [14]. This unit, now called the growing degree-day (GDD), is defined as the amount of heat an organism needs to accumulate to achieve full development. Although its main application remains in agriculture, the GDD concept has also been used for predicting the development of parasites [15], monitoring snail replication as well as the maturation of the schistosome sporocyst stages inside the snail [16,17].

The effect of temperature alterations with regard to the potential distribution of schistosomiasis has been addressed in China [17,18,19] and Africa [20,21], but a global picture of potential long-term transmission alterations ascribed to climate change is still missing. To address this question, we felt that it was warranted to theorize about the intermediate- and long-term global distribution possibilities of this disease considering possible future dispersion of the intermediate snail host into non-endemic areas. The IPCC focuses a lot on temperature predictions and this variable has a strong influence on snail survival. Importantly, even minor temperature increases play a role if they cover a sufficiently long period of the annual cycle. For these reasons, we used GDD as the key parameter to evaluate the potential, future transmission risk of schistosomiasis. 

## 2. Materials and Methods

When preparing the risk maps, we only considered the predicted temperature increase in relation to two required needs. These were the needs of the snail (three species) and that of the parasite stages inside the snail (mother sporocyst, daughter sporocyst and cercaria). Naturally, this approach was highly theoretical as many parameters other than temperature govern the potential distribution of schistosomiasis. However, it represents a first step in an investigation that can be developed further.

### 2.1. Study Area

Aiming to give a global overview of potential risk areas, a digital map of the world with country boundaries was downloaded from ArcGIS (ESRI, Redlands, CA, USA). Areas of particular risk for the change in schistosomiasis transmission were the current northern limits of the infection and slopes bordering high-altitude mountains. The southern limit was the southernmost tip of Africa and Chile and Brazil in South America as the sea rather than temperature limiting transmission at all other longitudes. 

### 2.2. Meteorological Model

Modelling future temperatures builds on four representative concentration pathways (RCPs) representing four greenhouse gas concentration (not emission) trajectories, which were adopted by the IPCC for its 5th assessment report (AR5) in 2014 [22]. These RCPs superseded the Special Report on Emissions Scenarios (SRES) projections published in 2000 and were previously used for this purpose. Unlike simpler models which make mixing assumptions regarding processes internal to a presumed cell, while other functions govern the interface between such cells, the GCMs describe four possible climate futures, which are all considered possible depending on how much greenhouse gases (GHGs) are emitted in the years to come (see Figure 1). The four RCPs, i.e., RCP2.6, RCP4.5, RCP6 and RCP8.5 are named after a possible range of ‘radiative’ forcing relative to pre-industrial values (+2.6, +4.5, +6.0, and +8.5 W/m^2^, respectively) [23]. 

The IPCC relies also on Coupled Model Intercomparison Project (CMIP) experiments, a collaborative series of experiments designed to improve our knowledge of the GCMs and their interactions. The most recently completed phase of the project is CMIP5, which includes more metadata describing model simulations than previous phases. The findings are summarized in AR5 (https://cmip.llnl.gov/cmip5/index.html). In this study, all models were based on CMIP5 with RCP4.5, which represented a medium GHG emission increase. The temperature prediction data used were derived from GCMs created in a grid format of 0.5 × 0.5 degrees latitude and longitude by five well-known research centers representing the most generally accepted view of the world’s major climate systems (Table 1). The datasets we used consisted of a baseline temperature based on the 1961–1990 period and predicted periods 2021–2050 and 2071–2100. These were based on an ensemble of the five predicted climate scenarios between 2006 and 2100 with special reference to the monthly mean temperature (T_mean_), the maximum temperature of the warmest months (T_max_) and the minimum temperature of the coldest months (T_min_), which are reminiscent of the BIO_1_, BIO_5_ and BIO_6_ variables of the ‘WorldClim’ dataset [24]. Thirty years of averaged monthly temperatures (T_mean_, T_max_ and T_min_) spanning the periods 2021–2050 and 2071–2100 were extracted.

### 2.3. Biological Model

The GDD, being the temperature above a critical threshold multiplied by the duration needed, can be articulated as T_ave_−T_base_, where the former is the average daily temperature and the latter the base temperature or the lowest temperature at which development of an organism can occur. The annual sum of GDD, termed AGDD, determines fairly well the potential for the spatial distribution of living organisms. As the level of the GDD required to complete the development of an organism is fairly constant, organisms with high heat unit requirements are more likely to develop into mature stages in areas where AGDD is high. We found this unit useful for use as a means to define the geographical limits of schistosomiasis transmission, mainly based on the climate impact on the snail intermediate host but also on its effect on the development of the sporocyst inside the snail.

With the aim of assessing the effect of temperature on schistosomiasis development, we created biological models with reference to the parasites by calculating potential transmission indices (PTIs) derived from GDD and AGDD. The GDDs for *S. japonicum*, *S. mansoni* and *S. haematobium* were 853, 268 and 298 degree-days, respectively [17,25,26] and the efficient temperature ranges ($T_{low}$ and $T_{high}$) for the development of *S. japonicum*, *S. mansoni* and *S. haematobium* were 18–35 °C, 16–35 °C and 18–32 °C, respectively [16,25,26]. The AGDD for each grid was calculated by summing up the difference between the mean monthly temperature ($T_{ij}$) at location *i* and month *j* and the efficient temperature range for parasite development: (1)$$AGDD_{i}=\sum_{j=1}^{12}(T_{ij}-T_{low})\times day_{j}\times 1(T_{low}<T_{ij}<T_{high}),$$
where $day_{j}$ indicates the number of days in month *j*, 1(…) is the indicator function, giving the value of 1 if the condition within the parenthesis is true, otherwise 0. 

The $PTI_{i}$ was calculated for each pixel grid *i* in the periods 1961–1990, 2021–2050 and 2071–2100, according to Equation (2). Only *PTI* values above 1 were considered to be of relevance (i.e., areas where schistosomiasis transmission could potentially occur).
(2)$$PTI_{i}=(AGDD_{i}/GDD_{s})\times 1(AGDD_{i}/GDD_{s}>1),$$

*Oncomelania*, the intermediate snail host of *S. japonicum*, can only survive if the January mean temperature is higher than 0 °C [17,18], while the most common *S. mansoni* snail host exists in areas where the yearly $T_{\max}$ is between 20–33 °C [27]. The corresponding optimal temperature range of the intermediate snail host for *S. haematobium* is 15.5–31 °C [28]. The vector survival threshold masks were overlapped on the PTI maps for the three different *Schistosoma* species investigated.

### 2.4. Geographical Information System (GIS) and Risk Assessment

We created a GIS database using ArcGIS, version 10.5 (ESRI; Redlands, CA, USA) for the investigation of baseline and change of transmission regions. 

#### 2.4.1. Baseline, Predicted Transmission Region and PTI

The PTI of the three main different schistosomiasis species for the periods 1961–1990, 2021–2050 and 2071–2100 were imported and presented as map overlays. On top of the PTI layer, we overlapped the thresholds for the intermediate hosts for the three species: 0 °C of the January $T_{mean}$ for *Oncomelania* (*S. japonicum*), yearly $T_{\max}$ between 20 °C and 33 °C for the *Biomphalaria* snail species (*S. mansoni*) and a $T_{\max}$ between 15.5 °C and 31 °C for the *Bulinus* species (*S. haematobium*).

#### 2.4.2. Change of Transmission Region, PTI due to Predicted Climate Change

We compared the spatial extent of the sensitive potential transmission region at two time periods: 2021–2050 vs. baseline 1961–1990 and 2071–2100 vs. baseline. We highlighted the change of PTI between the two time periods to present the transmission intensity difference due to the predicted climate change.

### 2.5. Model Validation by Ground Truth Data

*S. japonicum* was selected to evaluate the model prediction. All prediction results by pixel grids were compared with the same format data derived from the historic schistosomiasis transmission data provided by National Institute of Parasitic Diseases (NIPD), China Centers for Disease Control (CDC). The model sensitivity and specificity of the prediction results were evaluated by the following three equations:(3)$$SE=\frac{Pe}{Pte},$$
(4)$$SP=\frac{Pn}{Ptn},$$
(5)$$TM=\frac{Pen}{Pa},$$
where *SE* is sensitivity, *SP* is specificity, *TM* is the total agreement rate, Pe is the correctly predicted endemic pixels, Pte is all endemic pixels, Pn is the correctly predicted endemic pixels, Ptn is all non-endemic pixels, Pen is correctly predicted endemic and non-endemic pixels and Pa are all pixels.

## 3. Results

### 3.1. Change of Transmission Region and PTI for Schistosomiasis due to Climate Change

The average temperatures predicted by the IPCC for the two time periods 2021–2050 and 2071–2100, based on the average of the five GCM results, seemed increasingly likely in the event that efforts to curb the GHG emissions were not stepped up more than they were so far. Figure 2, Figure 3 and Figure 4 use IPCC’S predicted temperatures for the two future time periods 2021–2050 and 2071–2100 but rather than comparing with a preindustrial average, we used the 1961–1990 period for which we had available PTI data as a baseline. The blue color shown in the figures designated the ocean, while the background of each plot was based on the PTI baseline. 

For *S. japonicum*, we only applied the low temperature constraints. The global warming displayed heterogeneous patterns with respect to snail survival. For example, the current areas endemic for *S. japonicum* in the Sichuan Province were predicted to become unsuitable for snail habitats. On the other hand, the downstream regions between the Sichuan and Hunan/Hubei provinces corresponding to the area above the Three Gorges Dam, which is currently non-endemic, might well present a potential endemic risk in the future. The predicted potential transmission areas of *S. japonicum* moving northward was not so obvious in 2021–2050 as it might have become in the 2071–2100 period.

For *S. mansoni* and *S. haematobium*, in both timeframes forecast, the potential transmission areas could retreat from the Saharan areas as the desert might encroach on this part of the continent due to diminishing or discontinued rainfall. The future transmission areas of *S. mansoni* might shrink more than those of *S. haematobium*, while schistosomiasis in Ethiopia and other highlands in Africa could reach higher with warmer temperatures.

Suitability varies from 0 over the spectrum to the most likely (red). Grey areas signify areas that might potentially shrink as the temperature would become less suited to the specific snails for the different schistosomal species.

### 3.2. Model Validation by Ground Truth Data

The model prediction results were validated by the ground truth data in China. The sensitivity and specificity of model prediction were 0.78 and 0.89, respectively. The total prediction match rate was 0.88. 

## 4. Discussion

Three schistosome species pose the main infection risk for humans. *S. japonicum* is the single schistosome species in China, the Philippines and three closely situated small foci in Celebes, Indonesia. It is transmitted by *Oncomelania* snails, whose latitudinal movements over land during most of the last million years may have been prevented by the development of the Himalaya plateau serving as a barrier. The endemic areas for *S. mansoni* and *S. haematobium* overlap to a great extent on the African continent and in the Middle East, but not everywhere. The former, the most abundant of the schistosome species, has also found an exclusive niche in the Northeast part of South America and on some of the Caribbean islands but are now disappearing from the latter, however this is not due to climate change [29]. Its distribution is supported by the broad geographic range of *Biomphalaria* spp. snails that serve as obligatory hosts for its larval stages. Recently, however, *Bi. straminea* has been identified in the Hong Kong Special Administrative Region and Shenzhen City in the Guangdong Province of China and has remained there since the 1970s [30], a fact that could eventually result in transmission of *S. mansoni* in parallel with *S. japonicum* in China. *S. haematobium* is confined to Africa and the Middle East due to its reliance on *Bulinus* spp. intermediate snail hosts [31]. Although this schistosome species must also have been brought from Africa to Latin America during the slave trade of the 16th and 17th century, only *S. mansoni* became established there because of the absence of *Bulinus* snails. Although this information indicates that schistosome species and their specific snails can spread into various new areas, this is not of a magnitude that requires it to be considered when discussing the risk for schistosomiasis due to temperature augmentation. 

To assess the potential impact of climate change on the transmission of schistosomiasis in the world we developed a biological model using available data from the literature. Compared to *S. japonicum*, both *S. mansoni* and *S. haematobium* required less total heat energy to complete one generation [17,25,26]. Therefore, the transmission intensity of the two latter species is generally somewhat stronger than that of *S. japonicum*. However, the three different intermediate snail hosts had different temperature thresholds for survival that came into play in certain geographical areas. For example, the distribution of the *Oncomelania* snail, utilized by *S. japonicum*, was constrained by the temperature in January (when below 0 °C), which effectively limited its progress northwards, while *Biomphalaria* (*S. mansoni*) and *Bulinus* (*S. haematobium*) had thermal requirements that included an annual $T_{\max}$ of 15.5–31 °C and 20–33 °C respectively [27], which is one of the factors (so far) keeping these snails from entering Europe and other northern territories. Interestingly, *S. haematobium* appears to have a better chance of a foothold in Europe than *S. mansoni*, at least in the middle part of the century (Figure 3).

With respect to snail survival, the GDD approach in 2008 predicted an extension of 662,373 and 783,883 km^2^ potentially endemic areas for schistosomiasis japonica in China in 2030 and 2050, respectively [17]. Yang et al. noted that the 0–1 °C January isotherm of 1971–2000 had already shifted from 33°15′ N to 33°41′ N compared to the 1961–1990 baseline, expanding the area by 41,335 km^2^, resulting in an additional 20.7 million people at risk of schistosomiasis in China [32]. For *S. japonicum*, the predicted temperature increases based on our biological model and the average ensemble of the five GCM models would result in transmission starting to extend northward by 2050 (Figure 2a) adding a considerably larger margin by 2100 (Figure 2b). It is also conceivable that the transmission intensity will increase in areas already endemic for schistosomiasis.

It is important for the existence or absence of the parasite and vital for prediction model validation that more information is made available also for *S. mansoni* and *S. haematobium*. However, the change in the geographic transmission region for these two species due to climate change is more complex than in China. With temperatures increasing above $T_{\max}$, the historic transmission areas surrounding the Sahara region would retreat from the equator reducing the transmission regions of both species. However, the current risk areas would extend further into the northern or southern current marginal transmission areas, including the Mediterranean countries. Indeed, schistosomiasis hematobia has recently been identified in Corsica, France [33]. Although it is questionable whether this extension into Europe is due to climate change, rather large areas of southern Europe will become habitable for the snail hosts in the future (Figure 3 and Figure 4). Overall, however, the total transmission areas should be reduced. The changing transmission patterns in terms of region and intensity are heterogeneous and new populations at risk will probably appear. Further research is warranted to work out the geographical probabilities. 

A limitation of our current modelling approach is that it emphasizes the role of temperature but does not take into account the role of rainfall and the potential interaction between temperature and rainfall. It is difficult to say whether our model is conservative or whether these additional effects might further amplify the extent of changes predicted on the basis of temperature alone. Thus, recent improvements in modelling global trends in precipitation and water availability should become an integral part of present and future predictions of climate change and variability on infectious disease dynamics, including schistosomiasis. Recent years have seen widespread massive flooding, which can lead to snail dispersal and an epidemic outbreak of schistosomiasis if occurring in or close to endemic areas. In addition, the issue of altitude, that has been neglected so far, might under-evaluate changing transmission patterns in mountainous regions, which would have major implications for parts of China and many African countries. Additionally, most climate studies, including the current one, discuss large geographical scales, i.e., national, regional and global levels. To capture localized climate changes, regional climate models are needed that can assign meteorological parameters at relatively small scales also. Such climate models can provide information with useful local detail including realistic extreme events and generate detailed projections of the future climate. Any discussion of the future distribution of schistosomiasis, presumed to be induced by climate change would be futile without exact information of the prevailing distribution of the infection, both in humans and in the intermediate snail host. It is therefore critical that the diagnostic techniques applied are sensitive enough to find even the lightest infection. 

This work is based on the best possible models available today on the future climate for the rest of the century. Without a doubt, however, future increased computing power will permit more comprehensive simulations, better represented parameterized processes and more accurate projections at all levels, which might well result in slightly different projections. In addition, natural phenomena, such as cataclysmic volcanic eruptions like that which occurred in Krakatoa in 1883, major meteoric hits, etc., that would completely upend any prediction of the kind discussed here cannot be completely ruled out. Even without such events, we still expect further changes in the forthcoming 6th assessment report (AR6) scheduled for release in 2022.

## Figures and Tables

**Figure 1 tropicalmed-03-00117-f001:**
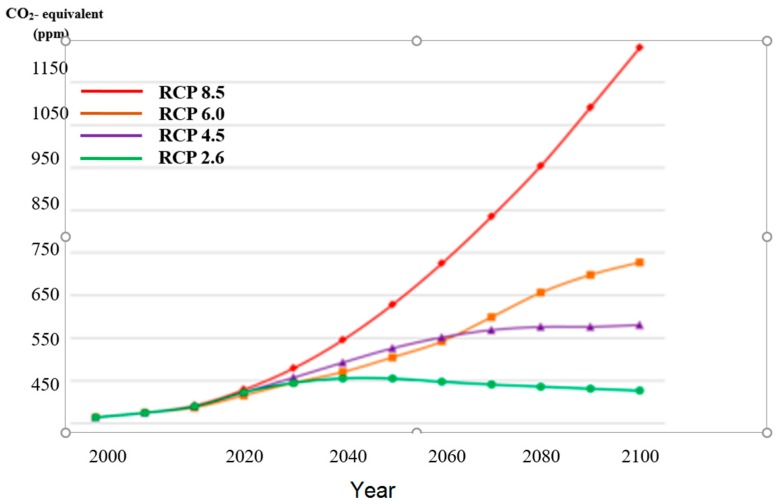
Estimated future equivalent atmospheric CO_2_ concentrations including all ‘forcing’ agents according to the four representative concentration pathways (RCPs).

**Figure 2 tropicalmed-03-00117-f002:**
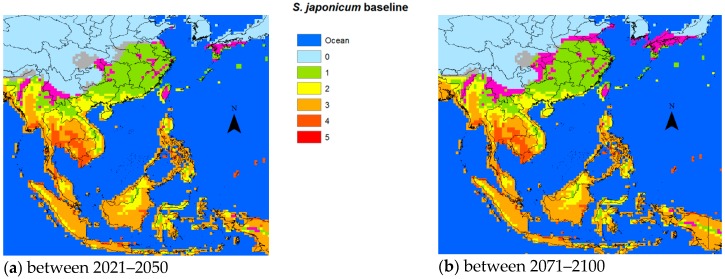
Change of risk area for *S. japonicum* vs. the baseline. (**a**) 2021–2050 vs. the baseline, (**b**) 2071–2050 vs. the baseline.

**Figure 3 tropicalmed-03-00117-f003:**
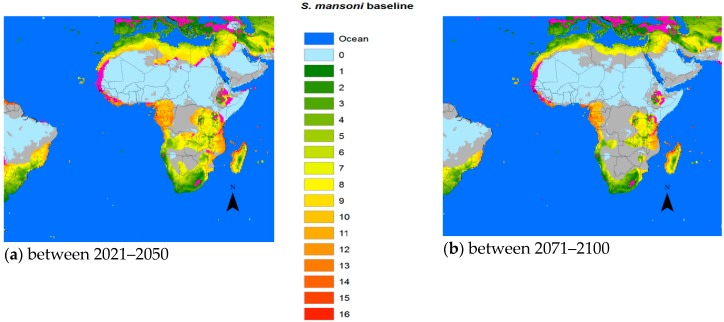
Change of risk area for *S. mansoni* vs. the baseline. (**a**) 2021–2050 vs. the baseline, (**b**) 2071–2050 vs. the baseline.

**Figure 4 tropicalmed-03-00117-f004:**
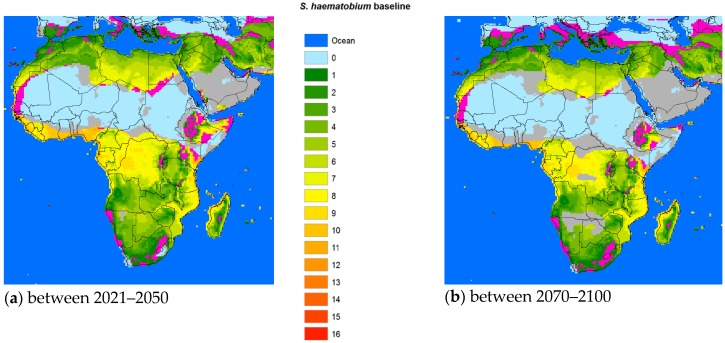
Change of risk area for *S. haematobium* vs. the baseline. (**a**) 2021–2050 vs. the baseline, (**b**) 2071–2050 vs. the baseline.

**Table 1 tropicalmed-03-00117-t001:** The five global circulation models used in this study.

GCM ^a^	Characteristics	Developing Centre
ACCE-SS1-0	Based on the UK MetOffice UM atmosphere model, the GFDL MOM4p1 ocean model, the LANL CICE4.1 sea-ice model and the MOSES2 land surface model.	The Australian Community Climate and Earth-System Simulator (ACCESS) weather models
IPSL-CM5A_LR	An atmosphere-land-ocean-sea ice model with representations of the carbon cycle; the stratospheric chemistry and the tropospheric aerosol chemistry	Institut Pierre Simon Laplace Climate Modelling Centre (IPSL-CMC) of Centre National de la Recherche Scientifique (CNRS); Paris; France
HadGEM2-AO	A configuration of the HadGEM2 model which is an atmosphere-only simulation with other component interfaces replaced with ancillary file input.	UK Met Office Hadley Centre
CanESM2	The second generation Canadian Earth System Model (CanESM2) consists of the physical coupled atmosphere-ocean model CanCM4 coupled to a terrestrial carbon model (CTEM) and an ocean carbon model (CMOC).	Canadian Centre for Climate Modelling and Analysis
GISS-E2-H-CC	Based on Earth system models that include interactive atmospheric chemistry, aerosols, carbon cycle and other tracers, as well as the standard atmosphere, ocean, sea ice and land surface components.	National Aeronautics and Space Administration (NASA) Goddard Institute for Space Studies

^a^ Global Circulation Model.

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
