# Peer review of "Potential Impact of Climate Change on Schistosomiasis: A Global Assessment Attempt"

_tropicalmed, 2018, doi:10.3390/tropicalmed3040117_

Round 1
Reviewer 1 Report
The paper presents a very important subject of climate change, how this will change snail distributions and the affect of potential schistosomiasis transmission. I have only very minot comments.
Line 30 - schistosome infections are not referred to a infecting abdominal cavities. Amend this to fit other publications.
Line 240 - S. haematobium is more abundant than S. mansoni
Author Response
Comments and Suggestions for Authors
The paper presents a very important subject of climate change, how this will change snail distributions and the affect of potential schistosomiasis transmission. I have only very minot comments.
Response: thank you for the positive feedback.
Line 30 - schistosome infections are not referred to a infecting abdominal cavities. Amend this to fit other publications.
Response: We replaced “abdominal capillaries” as “inferior mesenteric vein”. The sentence is written as now “Schistosomiasis is caused by trematode parasites with a predilection for the inferior mesenteric vein of the human definitive host. ”
Line 240 - S. haematobium is more abundant than S. mansoni
Response: Thank you for pointing out our mistake. We exchanged the sequence of S. haematobium and S. mansoni. The sentence has been corrected accordingly as “S. haematobium and S. mansoni overlap to a great extent…”
Reviewer 2 Report
Abstract
Line 14 – GHF should be defined here
Line 15 – assume this CHG should be GHG…
Introduction
Line 46 – I’m not sure this sentence makes sense. Schisto is not as neglected as other NTDs because it is quite prevalent? Would be better perhaps to state that while it is an NTD that there is still a large research focus on it as opposed to some of the other NTDs.
Line 48-52 – these two sentences could do with revision
Methods
Line 108 – What about water temp for hatching eggs? That seems to be an important consideration –at least if you try hatch them in a flask…they seem to have no problems in the wild!
Line 128 and 131/132 – Not sure if it’s just happened when converting to PDF, but ‘experiments’ on line 128 and ‘summarized in AR5’ line 132-132 seems to be in much larger font than the surrounding words.
Line 163 – wow, that’s quite a difference for japonicum compared to the other two species
Line 174 – Is the J on January a different font size?
Line 211 – although will schisto have been eliminated in China before this al becomes an issue...
Figure 1 does not appear to have a title.
Figures 2-4, what is the pink representative of? Not present in the legend
Figures not referred to in the results text – and Fig. 3 seems to be referred to first followed by Fig 2 in the discussion,Could do with a more extensive caption as well to explain what is happening – i.e. which models used for (a) vs (b)
What about suitable areas in Europe? There has been an outbreak of schistosomiasis in Europe relatively recently and movement of migrants from endemic areas may mean this becomes more common.
Boissier J, Moné H, Mitta G, Bargues MD, Molyneux D, Mas-Coma S. Schistosomiasis reaches Europe. The Lancet Infect Dis. 2015;15(7):757-8. doi: http://dx.doi.org/10.1016/S1473-3099(15)00084-5.
Line 245 – ‘It is transmitted by Oncomelania snails and the reason is speciation and geography, such as the development of the Himalaya plateau that may have served as a barrier preventing latitudinal moves over land during most of the last million years’ Needs rewriting – and the reason for what? The reason it is restricted to SEA? Or, if I read the following sentence, are you referring to the snail not the schisto species?
Line 249 – ‘China’ is in larger text
Line 251 – ‘Depending on its intermediate Bulinus spp. snail hosts, S. haematobium is confined to Africa and the Middle East [31].’ Maybe switch this around – S. haematobium is confined to Africa and the Middle East due to it’s reliance on Bulinus spp. intermediate snail hosts.
Line 254-257 – Why not? Isn’t the risk of schisto due to climate change due to moving into new areas?
Line 260 – only half of haematobium is in italics
Line 279 – delete ‘today’
Line 315 – 319 – This seems quite unnecessary, I would remove
Author Response
Abstract
Line 14 – GHF should be defined here; Line 15 – assume this CHG should be GHG…
Response: We have re-arranged the sequence and correct mistakes.
Introduction
Line 46 – I’m not sure this sentence makes sense. Schisto is not as neglected as other NTDs because it is quite prevalent? Would be better perhaps to state that while it is an NTD that there is still a large research focus on it as opposed to some of the other NTDs.
Response: we modified as “Schistosomiasis is not as neglected as many other tropical diseases since it has a large research focus and still remains one of the most prevalent infections in the world,…”
Methods
Line 108 – What about water temp for hatching eggs? That seems to be an important consideration –at least if you try hatch them in a flask…they seem to have no problems in the wild!
Response: Egg hatching can be self-adjusted within anytime during the day when temperature is suitable. For instance, if temperature at noon is too hot, the best hatching time will be in the morning or after sunset. In this study, we mainly focus on relatively longer term survival thermal conditions of snail hosts and parasites.
Line 128 and 131/132 – Not sure if it’s just happened when converting to PDF, but ‘experiments’ on line 128 and ‘summarized in AR5’ line 132-132 seems to be in much larger font than the surrounding words.
Response: Font has been corrected accordingly.
Line 163 – wow, that’s quite a difference for japonicum compared to the other two species
Response: Yes, S. japonicum requires more thermal quantity.
Line 174 – Is the J on January a different font size?
Response: They are the same font size.
Figure 1 does not appear to have a title.
Response: The title of Fig.1 is under the Figure. “Figure 1. Estimated future equivalent atmospheric C02 concentrations including all ‘forcing’ agents according to the four representative concentration pathways (RCPs).”
Figures 2-4, what is the pink representative of? Not present in the legend
Response: The description is in lines 231-234. It is written “'Suitability varies from 0 over the spectrum to the most likely (red). Grey areas signify areas that might potentially shrink as the temperature there would become less suited to the specific snails for the different schistosomal species'.
Figures not referred to in the results text – and Fig. 3 seems to be referred to first followed by Fig 2 in the discussion, Could do with a more extensive caption as well to explain what is happening – i.e. which models used for (a) vs (b)
Response: We swapped Fig. 3 and Fig.4 to be consistent with Methods and Results text. Models for (a) and (b) are the same model, except different time frames, 2021-2050 and 2071-2100.
What about suitable areas in Europe? There has been an outbreak of schistosomiasis in Europe relatively recently and movement of migrants from endemic areas may mean this becomes more common.
Boissier J, Moné H, Mitta G, Bargues MD, Molyneux D, Mas-Coma S. Schistosomiasis reaches Europe. The Lancet Infect Dis. 2015;15(7):757-8. doi: http://dx.doi.org/10.1016/S1473-3099(15)00084-5.
Response: In Discussion part, we already wrote a couple of lines “However, the current risk areas would extend further into the northern or southern current marginal transmission areas, including the Mediterranean countries. Indeed, schistosomiasis haematobia has recently been identified in Corsica, France [33]. Although it is questionable if this extension into Europe is due to climate change, rather large areas of southern Europe will become habitable for the snail hosts in the future (Figs 3 and 4).”
Line 245 – ‘It is transmitted by Oncomelania snails and the reason is speciation and geography, such as the development of the Himalaya plateau that may have served as a barrier preventing latitudinal moves over land during most of the last million years’ Needs rewriting – and the reason for what? The reason it is restricted to SEA? Or, if I read the following sentence, are you referring to the snail not the schisto species?
Response: It has been re-written as “It is transmitted by Oncomelania snails, whose latitudinal moves over land during most of the last million years may have been prevented by the development of the Himalaya plateau serving as a barrier.”
Line 249 – ‘China’ is in larger text
Response: The font has been corrected accordingly.
Line 251 – ‘Depending on its intermediate Bulinus spp. snail hosts, S. haematobium is confined to Africa and the Middle East [31].’ Maybe switch this around – S. haematobium is confined to Africa and the Middle East due to it’s reliance on Bulinus spp. intermediate snail hosts.
Response: Thank you for your suggestion. The sentence has been re-arranged.
Line 254-257 – Why not? Isn’t the risk of schisto due to climate change due to moving into new areas?
Line 260 – only half of haematobium is in italics
Response: The mistake has been corrected.
Line 279 – delete ‘today’
Response: It has been done.